# Genome-Wide Analysis and Characterization of the *SDR* Gene Superfamily in *Cinnamomum camphora* and Identification of Synthase for Eugenol Biosynthesis

**DOI:** 10.3390/ijms251810084

**Published:** 2024-09-19

**Authors:** Yueting Zhang, Chao Fu, Shifang Wen, Ting Zhang, Xindong Wang

**Affiliations:** 1Camphor Engineering and Technology Research Center of National Forestry and Grassland Administration, Jiangxi Academe of Forestry, Nanchang 330032, China; yuetingzhang918@163.com (Y.Z.); plantfuchao@163.com (C.F.); wenshifang@jxlky.cn (S.W.); zhangtycx@163.com (T.Z.); 2Jiangxi Provincial Key Laboratory of Improved Variety Breeding and Efficient Utilization of Native Tree Species (NO. 2024SSY04091), Jiangxi Academe of Forestry, Nanchang 330032, China

**Keywords:** *Cinnamomum camphora*, short-chain dehydrogenase/reductase (SDR) superfamily, expression profiles, eugenol synthase, SDR460A family

## Abstract

Short-chain dehydrogenase/reductases (SDRs) are the largest NAD(H)-dependent oxidoreductase superfamilies and are involved in diverse metabolisms. This study presents a comprehensive genomic analysis of the *SDR* superfamily in *Cinnamomum camphora,* a species that is one of the most significant woody essential oil plants in southern China. We identify a total of 222 CcSDR proteins and classify them into five types based on their cofactor-binding and active sites: ‘atypical’, ‘classic’, ‘divergent’, ‘extended’, and ‘unknown’. Phylogenetic analysis reveals three evolutionary branches within the CcSDR proteins, and further categorization using the SDR-initiative Hidden Markov model resulted in 46 families, with the CcSDR110C, CcSDR108E, and CcSDR460A families being the most populous. Collinearity analysis identified 34 pairs of Cc*SDR* paralogs in *C. camphora*, 141 pairs of *SDR* orthologs between *C. camphora* and *Populus trichocarpa*, and 59 pairs between *C. camphora* and *Oryza sativa*. Expression profile analysis indicates a preference for the expression of 77 *CcSDR* genes in specific organs such as flowers, bark, twigs, roots, leaves, or fruits. Moreover, 77 genes exhibit differential expression patterns during the four developmental stages of leaves, while 130 genes show variance across the five developmental stages of fruits. Additionally, to explore the biosynthetic mechanism of methyl eugenol, a key component of the leaf essential oil in the methyl eugenol chemotype, this study also identifies eugenol synthase (EGS) within the CcSDR460A family through an integrated strategy. Real-time quantitative PCR analysis demonstrates that the expression of *CcEGS* in the leaves of the methyl eugenol chemotype is more than fourfold higher compared to other chemotypes. When heterologously expressed in *Escherichia coli*, it catalyzes the conversion of coniferyl acetate into a mixture predominantly composed of eugenol (71.44%) and isoeugenol (21.35%). These insights pave the way for future research into the functional diversity of *CcSDR* genes, with a focus on secondary metabolism.

## 1. Introduction

The superfamily of short-chain dehydrogenases/reductases (SDRs) is one of the most ancient and includes a vast array of proteins from all domains of life [1,2]. Characteristic of their structures are Rossmann folds, which typically feature a central β-sheet flanked by two or three α-helices [3]. These proteins also exhibit conserved structural motifs: a coenzyme-binding site defined by the sequence TGxxx[AG]xG or GxxGxxG in the N-terminal region and an active site with the motif YxxxK in the central region [4,5]. The substrate-binding motif, which is structurally variable, is located in the C-terminal region, forming a cleft adjacent to the coenzyme-binding site in the three-dimensional structure [6]. Based on these structural features and sequence motifs, the SDR superfamily is categorized into seven types: classical (C), extended (E), complex (X), divergent (D), atypical (A), intermediate (I), and unassigned (U) [3]. The C type, with the most genes, usually consists of about 250 amino acid residues. The E type, with the second-largest number of proteins, features an additional 100-residue domain in the C-terminal region, expanding the substrate-binding site significantly compared to classical SDRs.

Despite low sequence identities ranging from 15 to 30% among SDRs, the identification of members within each type primarily relies on annotations from multiple Pfam entries, such as PF00106, PF01370, PF01073, PF13561, and PF08659 [3]. To enhance the understanding of SDR functions and to establish a sustainable and scalable nomenclature system, a nomenclature initiative has been developed using an HMM-based approach [7]. This system now lists 449 families on the SDR Enzymes webpage (accessed on 1 June 2024, http://www.sdr-enzymes.org), which serves as a comprehensive resource for the *SDR* gene superfamily [6].

The previous literature has documented the critical roles of SDRs in plant development and primary metabolism. For instance, enzymes such as UDP-glucose 4-epimerase 3 (AtUGE3) from *Arabidopsis thaliana*, along with AtUGE4 and OsUGE from *Oryza sativa*, are part of the SDR1E family and are pivotal in pollen development [8,9]. Additionally, the AtRHM2 enzyme (SDR2E) was found to be essential for the normal development of the seed coat epidermal layer [10]. These proteins are involved in carbohydrate catabolism within the cell wall. Proteins from the SDR73C and SDR98U families participate in the biosynthesis of chlorophyll and its intermediates, with low expression levels leading to changes in leaf color [11,12,13,14]. Similarly, SDR7C, SDR81U, and SDR83U proteins are key components in chloroplast biogenesis, and are essential for the normal growth and development of plants [15,16,17]. Furthermore, SDRs such as those from the SDR12C, SDR87D, SDR117E, and SDR152C families are involved in lipid catabolism and are crucial for normal plant growth. Mutations in genes like *AtKCR1* (SDR12C) and *AtFARs* (SDR117E) cause embryo lethality and abnormal pollen development, respectively [18,19]. Mutations in *AtKAS I* (SDR152C) and *AtMod1* (SDR87D) both result in multiple morphological defects, including chlorotic and curly leaves, reduced fertility, and semidwarfism [20,21].

The functional diversity of plant SDRs in secondary metabolism has been extensively reviewed by Tonfack et al. [22]. For instance, proteins within the SDR460A and SDR110C families were shown to be involved in phenylpropanoid and terpenoid metabolism, respectively. Phenylpropene synthases, part of the SDR460A family, participate in the biosynthesis of various phenylpropenes and, along with pinoresinol–lariciresinol reductase (PLR), isoflavone reductase (IFR), and phenylcoumaran benzylic ether reductase (PCBER), form four distinct branches in the evolutionary tree of SDR460A, also known as the PIP reductase family [23,24,25]. In recent years, two types of phenylpropene synthases, eugenol synthase (EGS) and isoeugenol synthase (IGS), have been identified in various species [26,27,28,29,30,31,32,33]. These enzymes primarily produce eugenol or isoeugenol, utilizing coniferyl acetate and NADPH as substrates, respectively. In the SDR110C family, enzymes such as (−)-trans-isopiperitenol dehydrogenase from peppermint and (−)-trans-carveol dehydrogenase from spearmint convert (−)-trans-isopiperitenol to (−)-isopiperitenone and (−)-trans-carveol to (−)-carvone, respectively [34]. Homologous enzymes from the Lamiaceae family are capable of utilizing both trans- and cis-(−)-isopiperitenol and trans- and cis-(−)-carveol to produce the corresponding ketones isopiperitenone and carvone [35]. In *Wurfbainia villosa*, four borneol dehydrogenases from the SDR110C family selectively catalyze different configurations of borneol to generate the corresponding configurations of camphor [36].

Currently, there are limited investigations into the diversity of *SDR* genes in plants. Previous reports have documented the number of *SDR* genes in various plant species, including 142 in *Selaginella moellendorffii*, 178 in *A. thaliana*, 268 in *Populus trichocarpa*, 205 in *Vitis vinifera*, 315 in *Glycine max*, 227 in *O. sativa*, 230 in *Zea mays*, 237 in *Sorghum bicolor*, and 213 in *Medicago truncatula* [37,38]. To date, five types (C, E, A, D, and U) of SDRs have been identified in plants [37]. *C. camphora* is widely cultivated in southern China for its valuable timber and essential oil production. A multitude of terpenoids and phenylpropanoids from the essential oils of different chemotypes have been identified [39]. Understanding the biosynthesis of key compounds and the mechanisms underlying the formation of various chemotypes are crucial for the breeding and enhancement of camphor trees. Recently, three SDRs from the SDR110C family in *C. camphora* were described which were capable of converting borneol to camphor in vitro [40]. However, the genes responsible for the biosynthesis of key phenylpropanoids, such as methyl eugenol and eugenol in *C. camphora,* were covered. These compounds are extensively utilized in medical applications and as ingredients in insecticides. This work aims to identify the *SDR* gene superfamily of *C. camphora* at a genome-wide level, to provide expression information in various organs, and to characterize the biochemical function of CcEGS in the biosynthesis of eugenol. This investigation will offer valuable reference information for further elucidating the diverse roles of SDRs and provide new insights into the molecular basis of chemodiversity in *C. camphora*.

## 2. Results

### 2.1. Identification and Classification of CcSDR Genes

A total of 217 *CcSDR* genes in the genome of the camphor tree, in addition to an additional 5 *CcSDR* genes in the leaf transcriptome, were identified by PacBio sequencing, according to annotations by the HMM profile of SDRs and multiple Pfam entries (Pfam: PF00106, PF01370, PF01073, PF13460, and PF13561) (Appendix A). The lengths of the 222 CcSDR proteins varied, with CcSDR84C being the shortest at 249 amino acids and CcSDR2E3 being the longest at 678 amino acids. The isoelectric points of the CcSDR proteins ranged from 4.79 for CcSDR108E11 to 9.92 for CcSDR50E1 (Appendix A). Further classification divided the genes into five types: C (108 genes), E (77 genes), D (3 genes), A (22 genes), and U (11 genes). Ultimately, 217 genes were divided into 46 families, with CcSDR110C, CcSDR108E, and CcSDR460A being the top three families, comprising 36, 34, and 22 genes, respectively. Sixteen other families, such as SDR25C, 34C, and 40C, each contained only one protein. The remaining five genes (Cca.gene7945, Cca.gene13781, Cca.gene19321, Cca.gene25898, and Cca.gene37092), which had low HMM scores, were classified into the unclassified family.

### 2.2. Chromosome Localization and Duplication Analysis of CcSDR Genes

Chromosome localization analysis revealed that 215 *CcSDR* genes were unevenly distributed across twelve chromosomes (Chrs) (Appendix A). Chr01 harbored the highest number with 42 *CcSDR* genes, while Chr09 had the fewest with only 4 *CcSDR* genes. In addition, CcSDR65C9 and 65C10P were located on scaffold93 (Appendix A). Further gene structure analysis indicated that 217 genes contained varying numbers of exons, with 17 members having only a single exon and *CcSDR544C* having the highest count with 18 exons (Appendix A). A significant majority of members within the same family shared a consistent exon–intron structure. For example, 81.81% of the genes in the SDR460A family, 82.35% in the SDR110C family, and 69.70% in the SDR108E family were composed of five, two, and six exons, respectively (Appendix A).

Within the genome of *C. camphora*, 34 pairs of genes were identified as paralogous in the *CcSDR* gene superfamily, attributed to segmental duplications (Figure 1 and Appendix A). These *CcSDR* genes originated from 18 CcSDR families across five types, with some members having undergone more than one duplication event. To investigate the evolutionary dynamics of the CcSDR superfamily, the synonymous substitution rate (Ks), non-synonymous substitution rate (Ka), and Ka/Ks ratios were calculated for these 34 paralogous gene pairs. As shown in Appendix A, all gene pairs had Ka/Ks values below 1, suggesting purifying selection occurred in the CcSDR subfamily. Additionally, 101 genes were found to be arranged in tandem arrays across 32 chromosomal regions due to tandem duplication. For instance, *CcSDR108E1*~*108E8*, *CcSDR117E4*~*117E6*, and *CcSDR110C20*~*110C25* were located in the 84.50–85.60 Mb interval of Chr01, the 50.17–50.40 Mb interval of Chr03, and the 53.47–53.53 Mb interval of Chr06, respectively (Appendix A).

### 2.3. Collinearity Analysis of CcSDR Genes

Synteny analyses were conducted to investigate the evolutionary relationships of *SDR* genes between *C. camphora* and other species, including both from dicotyledonous and monocotyledonous plants. This analysis led to the identification of a total of 101 pairs of *SDR* orthologous genes between *C. camphora* and *P. trichocarpa* (Figure 2A), as well as 59 pairs between *C. camphora* and *Oryza sativa* (Figure 2B), respectively (Appendix A). Among these, 28 genes from 20 *CcSDR* gene families were found to be syntenic across the three species, with three proteins potentially involved in lipid metabolism and nine in glycometabolism. The results suggested that these proteins might play conserved and crucial roles throughout the evolution of terrestrial plants. However, almost all *SDR* orthologous genes from the CcSDR110C, 114C, and 108E families, which might participate in secondary metabolic processes and showed synteny between *C. camphora* and *P. trichocarpa*, were found to be non-syntenic between *C. camphora* and *O. sativa*. This finding implied that some SDRs involved in secondary metabolic processes had diverged following the separation of monocotyledonous and dicotyledonous plants.

### 2.4. Conserved Motifs in CcSDR Genes

Protein sequence analysis of the 222 CcSDRs revealed ten conserved motifs, ranging in size from 15 to 50 amino acid residues in width, designated as motifs 1 to 10 (Figure 3 and Appendix A). Motif 1, which included the cofactor binding site, was found in 208 CcSDR proteins. The remaining ten CcSDR proteins lacked the conserved motif 1 but possessed variants of the cofactor binding site. Motif 2, containing the active site, was present in 103 CcSDR proteins from the C and D types. However, the active site and its variants were found in all proteins from the C, E, and D types. Motifs 3, 4, 6, and 7 were primarily distributed among proteins of the C and D types. Additionally, motifs 5, 9, and 10 were exclusively found in the SDR460A and SDR110C families, respectively (Appendix A).

### 2.5. Phylogenetic Relationships of CcSDR Proteins

Based on the phylogenetic tree, the 222 CcSDR proteins were classified into a classical SDR fold with 111 members, an atypical SDR fold with 22 members, and an extended SDR fold with 89 members (Figure 4 and Appendix A). Moreover, CcSDR proteins in the classical SDR fold were further divided into three subgroups: clusters C1, C2, and C3, consisting of 52, 44, and 15 members from the C type, D type, and the unclassified family, respectively. All members of the CcSDR87D family were clustered into cluster C2. Similarly, CcSDR proteins in the extended SDR fold were also classified into three subgroups: clusters E1, E2, and E3, consisting of 35, 35, and 19 members from the E and U types, respectively. The phylogenetic topology suggested a closer relationship between the C and D types in the camphor tree, whereas a closer relationship was observed between the E and U types.

### 2.6. Expression Profile of CcSDR Genes in Different Organs

RNA sequencing (RNA-seq) was performed on flowers, bark, twigs, roots, leaves, and fruits to analyze the expression profiles of *CcSDR* genes in the different organs of the camphor tree. This process generated a total of 145.51 Gb of clean data with high-quality Q20 values ranging from 97.65% to 98.15%. Based on FPKM (fragments per kilobase of transcript per million mapped reads) values, 177 *CcSDR* genes from 46 families were found to be expressed in at least one organ (FPKM value > 1). With the exception of *CcSDR17C2*, *25C*, *110C33*, *544C*, *6E4*, *371E*, and *370U*, the remaining 96.05% (170) *CcSDR* genes showed differential expression across the six organs (Figure 5, Appendix A). Seventy-eight *CcSDR* genes exhibited significantly higher expression in specific organs compared to the other five, including sixteen genes in flowers, eleven in barks, four in twigs, ten in roots, twenty-three in leaves, and sixteen in fruits (Figure 5). Notably, 13 genes displayed organ-specific expression. For instance, *CcSDR460A6*, *460A13*, *460A14*, *110C13*, and *117E2* were specifically expressed in flowers, while *CcSDR57C2*, *119C2*, and *87D3* were exclusively expressed in fruits. Additionally, *CcSDR110C11*, *110C12*, and *110C25* were found to be expressed only in leaves. These genes were likely to be associated with specific organ development or the biosynthesis of particular metabolites.

### 2.7. Expression Profiles of CcSDR Genes during Leaf Development

RNA-seq of four developmental stages (S1, S2, S3, and S4) of leaves yielded 75.22 Gb of clean data (Figure 6A). To explore the possible roles of *CcSDR* genes during leaf development, we analyzed the expression patterns of *CcSDR* genes across these four stages. The FPKM values of 120 *CcSDR* genes exceeded one, and 77 genes were identified as differentially expressed genes (DEGs) among S1, S2, S3, and S4. There were 25, 18, 17, and 17 DEGs with peak expressions in S1, S2, S3, and S4 (Appendix A), respectively. Especially, five genes, namely *CcSDR7C4*, *12C1*, *12C2*, *110C21*, and *108E33*, exhibited specific expression in S1 and S2.

Furthermore, five distinct expression patterns of DEGs were identified through cluster analysis (Figure 6B). Specifically, 18 DEGs in clusters 1 and 3 had lower expressions in S1 and S2 than in S3 and S4, with a rapid increase from S2 to S3. In contrast, 21 DEGs in clusters 5 and 9 had higher expressions in S1 and S2 than in S3 and S4, with a significant decline from S2 to S3. In cluster 2, six DEGs exhibited significantly higher expression in S2, S3, and S4 relative to S1. DEGs from clusters 1, 2, and 3 might be positively associated with leaf growth. In clusters 4 and 8, ten DEGs accumulated the highest expression levels in S3, with five genes (*CcSDR460A21*, *2E1*, *6E2*, *367E3*, and *75U3*) showing more than twofold expression levels in S3 compared to S1, S2, and S4. The expression levels of the remaining 22 DEGs from clusters 6 and 7 gradually declined throughout the four developmental stages, suggesting they might have no effect or negative regulatory roles on leaf growth.

### 2.8. Expression Profiles of CcSDRs Genes during Fruit Development

According to the data from RNA-seq, a total of 148 genes from the *CcSDR* gene superfamily were transcribed (with FPKM values ≥ 1.00), and 88.59% (132) of them were identified as DEGs across the five developmental stages (S1, S2, S3, S4, and S5) of fruits (Figure 7A and Appendix A). Specifically, 51 DEGs exhibited peak expression in S1, followed by 31 DEGs in S2, 31 DEGs in S3, and 19 DEGs in S5. Overall, most of the genes (85.61%) exhibited elevated expression levels during the expanding growth stages of fruits (S1, S2, and S3), with a notable decrease in expressions upon reaching the ripening stage (S5). Moreover, 24 DEGs in S1, 10 DEGs in S2, 8 DEGs in S3, and 9 DEGs in S5 were found to be transcribed preferentially.

Additionally, cluster analysis revealed five distinct expression patterns of DEGs throughout the developmental stages (Figure 7B). *CcSDR* genes with peak expressions in S1, S2, S3, and S5 were grouped into clusters 4/7, 9, 1/2, and 3, respectively. The *CcSDR* genes from clusters 1 and 2 were likely associated with the rapid expansion of fruits (S1 to S3) or the synthesis of specific metabolites during these stages. Meanwhile, the *CcSDR* genes from cluster 3 could be implicated in fruit ripening. No *CcSDR* genes with peak expression in S4 were identified. The remaining genes, which showed fluctuating expression levels from S1 to S4, reached their lowest expressions in S5 and were categorized into clusters 5, 6, and 8, containing 13, 16, and 17 genes, respectively. These *CcSDR* genes might either have no influence on fruit ripening or could be involved in its negative regulation.

Considering that the seeds of the camphor tree are rich in medium-chain triglycerides [41], the expression profiles of gene from the SDR87D [21] and SDR152C [20] families that had been reported to be involved in de novo fatty acid biosynthesis were of our interest. Within the CcSDR87D family, *CcSDR87D1* exhibited near-constitutive expression patterns across different organs, with the expression levels in leaves and fruits being approximately twice as high as those in flowers. Concurrently, the expression of *CcSDR87D1* was more pronounced during the expansion phases of fruits (S1 to S3) than during the maturation phases (S4 to S5). *CcSDR87D2* was expressed at low levels only in fruits. *CcSDR87D3* showed preferential expression in S3 of fruits, with low-level expressions observed in both leaves and flowers, and it was almost not expressed in twigs. In the CcSDR152C family, *CcSDR152C2* also showed preferential expression in S3 of fruits, while *CcSDR152C1* was constitutively expressed across six organs (Figure 7B and Appendix A).

### 2.9. Identifying and Functional Characterization of CcEGS

We investigated and compared fresh leaf essential oil (LEO) profiles among 12 individuals from four chemotypes: the methyl eugenol, linalool, borneol, and camphor types (Figure 8A). The yields of LEOs (mL/g) for the methyl eugenol, linalool, borneol, and camphor types were 1.03~1.45%, 1.53~2.07%, 1.56~2.34%, and 1.47~2.12%, respectively. The predominant components were methyleugenol (40.38~54.69%), (−)-linalool (88.56~94.67%), (+)-borneol (73.24~81.45%), and camphor (55.96~70.54%), correspondingly. In the LEOs of the methyleugenol type, safrole (7.94~10.23%) and methyl isoeugenol (11.12~13.09%) were also detected, with phenylpropenes being the dominant components overall. In contrast, terpenoids were the major constituents of LEOs from the other three chemotypes. Relative contents of compounds such as α-caryophyllene, β-caryophyllene, 1,8-cineole, α-pinene, β-pinene, germacrene D, camphene, β-myrcene, α-terpineol, D-limonene, and terpinen-4-ol, which were more than 3%, were also found in one or more of the three chemotypes.

In previous reports, EGS and IGS were characterized as catalyzing the committed step in methyl eugenol and methyl isoeugenol biosynthesis in other plants, respectively, and identified as part of the PIP reductase family (Appendix A). To screen for candidate genes encoding EGS and IGS in *C. camphora* for the first time, a phylogenetic tree of the PIP family was constructed using biochemically characterized proteins from other plants (Appendix A). As shown in Appendix A, twenty-two CcSDR460A proteins were clustered into four clades, with particular attention given to the EGS/IGS clade containing CcSDR460A1, 460A2, 460A3P, 460A17, and 460A18, and the PCBER/EGS/IGS clade containing CcSDR460A10, 460A11P, 460A12, 460A13, and 460A14. These proteins were potential EGS or IGS candidates in the methyl eugenol and methyl isoeugenol biosynthesis pathways of the camphor tree.

Subsequently, the expression levels of these eight candidate genes, excluding two pseudogenes (*CcSDR460A3P* and *460A11P*), were tested in leaves of the methyl eugenol type, linalool type, borneol type, and camphor type, using quantitative real-time PCR (qRT-PCR) to identify *EGS* and *IGS* homologs in the camphor tree (Appendix A). Only *CcSDR460A1* displayed expression levels in the leaves of the methyl eugenol type that were more than twofold higher compared to the other three chemotypes (Figure 8B). Therefore, *CcSDR460A1* was selected as the candidate gene encoding EGS or IGS for subsequent molecular cloning and functional characterization.

The full-length coding sequence (CDS) of *CcSDR460A1* was successfully cloned from the methyl eugenol chemotype, as well as from three other chemotypes. The CDS sequences from these four chemotypes were found to be consistent. The open reading frame (ORF) of *CcSDR460A1* was identified to be 954 base pairs in length. Multiple sequence alignments revealed the presence of an NADPH-binding domain (GATGYLG) in the N-terminus of CcSDR460A1, specifically from Gly11 to Gly17 (Appendix A). CcSDR460A1 showed sequence identities of approximately 57.83%, 55.24%, 47.71%, 47.39%, 66.98%, and 56.51% with ObEGS1, CbEGS1, PhEGS1, CbEGS2, PhIGS1, and CbIGS1, respectively. To characterize the enzyme activity of CcSDR460A1, recombinant plasmids containing *CcSDR460A1* were transformed into Escherichia coli, and coniferyl alcohol was added to the culture medium as a substrate. Coniferyl alcohol can be converted to coniferyl acetate by endogenous acyltransferases in *E. coli* [26]. As shown in Figure 8C, *E. coli* harboring the recombinant plasmids produced a mixture in the medium, mainly consisting of eugenol (71.44 ± 2.13%) and isoeugenol (21.35 ± 1.69%) according to the results from gas chromatography–mass spectrometry (GC-MS). In contrast, *E. coli* containing the empty vector did not mediate the formation of eugenol and isoeugenol. These results confirmed that CcSDR460A1 had dual functionality but primarily acted as an EGS. Accordingly, SDR460A1 was identified as a CcEGS.

We also further investigated the subcellular localization of CcEGS by performing a transient expression assay in *Nicotiana benthamiana* leaves. The ORF of *CcEGS* was fused to the 5′-terminus of GFP (green fluorescent protein) and driven by the cauliflower mosaic virus (CaMV) 35S promoter for expression. The recombinant plasmids were transformed into *N. benthamiana* leaves via Agrobacterium-mediated transformation. As shown in Figure 8D, CcEGS-GFP fluorescence was observed in the cytosol, indicating that CcEGS was localized to cytosol. This experimental result corroborated the predictions made using the WoLF PSORT Server 0.2 and Softberry ProtComp 9.0 software, which also suggested a cytosolic localization for the protein.

## 3. Discussion

In this study, a total of 222 *SDR* genes were identified in the genome of *C. camphora*. Compared to the SDR superfamilies from other plants, the number of *CcSDR* genes was roughly similar to those from rice (227 genes), sorghum (237 genes), and maize (230 genes), and was lower than those from soybean (315 genes) and poplar (268 genes), but higher than those from Arabidopsis (178 genes) and grape (205 genes) [37]. Notably, both the CcSDR110C family and the CcSDR460A family expanded significantly, with 36 and 22 members, respectively, which seemed to be related to the abundance of secondary metabolites in camphor trees. The SDR110C family also expanded significantly in *W. villosa*, which was also rich in terpenoids [36]. Interestingly, 61.11% of *CcSDR110C* genes and 77.27% of *CcSDR460A* genes were produced by tandem duplication, suggesting that tandem duplication played an important role in the expansion of the two CcSDR families. However, no tandem duplication was observed in the SDR subfamily of *M. truncatula* [38]. Additionally, 28 *SDR* genes were identified as syntenic genes in poplar, rice, and camphor trees, with 12 of them involved in glycometabolism and lipid metabolism, implying that these *SDR* genes were likely essential for the growth of land plants.

In our finding, 78 *CcSDR* genes were preferentially expressed in different organs, and some homologous genes of them, as mentioned earlier, were involved in regulating the developmental processes of various organs. In the CcSDR1E family, *CcSDR1E2*, a homolog of *AtUGE4* [8], exhibited high expression levels across all organs, with particularly notable expression in flowers and during the fruit expansion phase. This pattern of expression implies that *CcSDR1E2* could play a pivotal role in both pollen development and the regulation of early fruit development. Genes from the SDR73C, SDR81U, SDR83U, and SDR98U families had been reported to participate in the biosynthesis of chlorophyll or in chloroplast biogenesis [11,12,13,14,16,17]. In the camphor tree, their homologs exhibited similar expression patterns, being preferentially expressed in leaves and highly expressed throughout the entire developmental period of the leaves. These findings suggested that they played crucial roles in the normal growth and development of leaves. AtVEP1 (SDR75U) was associated with reduced vascular development and the mutant was described as possessing an abnormal leaf venation pattern as well as thinner stems and roots [42]. In the genome of the camphor tree, CcSDR75U1, 75U2, 75U3, and 75U4 were all homologous to AtVEP1 with approximately 72% sequence identities but exhibited distinct expression patterns. The expression level of *CcSDR75U1* increased significantly during leaf development, while *CcSDR75U2*, *75U3*, and *75U4* were all preferentially expressed in roots, implying that both function divergence and redundancy of *SDR75U* genes occur in different organs of the camphor tree. AtHSD1 from the SDR119C family played an important role during seed maturation [43]. Despite a sequence homology of only 42% existing between CcSDR119C2 and AtHSD1, its expression significantly increased during the later stages of fruit development (S3–S5, cluster 3), suggesting that it might also be involved in regulating the maturation of camphor tree seeds.

Genes from the SDR87D and SDR152C families encoded chloroplastic enoyl-ACP reductase and chloroplastic 3-oxoacyl-ACP reductase, respectively. These two proteins are important components of the type II fatty acid synthase complex [20,21]. Based on the RNA-Seq data of six organs, CcSDR87D3 and CcSDR152C1 were inferred to be primarily responsible for fatty acid synthesis in seeds. Moreover, both *CcSDR87D3* and *CcSDR152C1* exhibited significantly higher expressions in stage 3 of fruits compared to other stages, suggesting that the biosynthesis of medium-chain triglycerides was particularly active in the developmental stage 3 of camphor tree fruits.

In camphor trees, key terpene synthases (TPS) such as CcTPS16, CcTPS28, CcTPS54, and CcTPS77 primarily mediated the formation of chemotypes with terpenoid compounds as the main components at the transcriptional level, including the 1,8-cineole type, linalool type, nerolidol type, and camphor (borneol) type [44]. However, the mechanism underlying the formation of chemotypes dominated by phenylpropanoid compounds remains currently unknown. In this study, the integrated analysis of leaf essential oil profiles and qRT-PCR results showed that the content of phenylpropanoid compounds (such as methyl eugenol, methyl isoeugenol, and safrole) were significantly positively correlated with the transcription levels of *CcEGS*. Similar findings had also been reported for *Melaleuca bracteata* [45]. These results indicated that the transcriptional level of *EGS* also played a significant regulatory role in the formation of methyl eugenol chemotypes in plants. In *E. coli* feeding assays, CcEGS was capable of catalyzing the conversion of coniferyl acetate into eugenol and a smaller amount of isoeugenol. The resulting product profile was similar to that observed with GdEGS [27], FaEGS2 [29], and DcE(I)GS1 [30]. In phylogenetic relationships, two distinct protein lineages of EGS had been identified in plants [46]. In addition to CcEGS, there were ten CcSDR460A genes included in the EGS/IGS clade and the PCBER/EGS/IGS clade, and it remains to be further clarified whether they could also convert coniferyl acetate into eugenol or isoeugenol. Additionally, the substrate specificity of CcEGS also needs further confirmation, considering that some EGSs derived from *Ocimum* can also utilize coumaryl acetate to produce chavicol [31]. In summary, based on the integration of essential oil phenotypes, transcription levels, and product profiles, CcEGS was considered to be the key enzyme mediating the synthesis of eugenol in camphor trees.

## 4. Materials and Methods

### 4.1. Plant Materials and RNA Sequencing

All experimental materials in this project were sampled from camphor trees cultivated at the Jiangxi Academy of Forestry, Nanchang City, China. Total RNA was extracted and purified following the instructions of the Quick RNA Isolation Kit (Huayueyang Biotech, Beijing, China). Five fresh organs—flowers, bark, twigs, roots, and leaves—were sampled from 10-year-old camphor trees in March 2023. Fresh leaves at four different developmental stages were collected from three clonally propagated seedlings in March, April, May, and August 2023. Five developmental stages of fruits were gathered from three 15-year-old trees in July, August, September, October, and November 2023. A total of 45 libraries, with three biological replicates each, were constructed and sequenced on an Illumina NovaSeq 6000 platform at Majorbio (Shanghai, China) and Allwegene (Nanjing, China).

### 4.2. Bioinformatics Analysis of CcSDR Genes

Based on the genome data and PacBio full-length transcriptome data of *C. camphora* published previously [44], the HMMER 3.2.1 software was utilized to identify candidate CcSDR proteins using five Pfam Hidden Markov Models (HMMs): PF00106, PF01073, PF01370, PF13460, and PF13561 (accessed on 1 June 2024, http://pfam.xfam.org/). Subsequently, redundant sequences, incomplete sequences, and possible sequencing errors were manually checked and excluded. The decision rules for the SDR inventory were adopted as described by Moummou et al. [37]. Ultimately, the types and families of CcSDRs were defined according to the SDR-type HMM and nomenclature initiative of HMMs [3,37].

The amino acid residues, molecular weight (MW), and isoelectric point of CcSDR proteins were analyzed using the online tool ProtParam (http://web.expasy.org/protparam/). The online tool GSDS 2.0 (https://gsds.gao-lab.org/) and the MEME suite 5.5.5 (https://meme-suite.org/meme) were employed to determine the exon–intron structure of *CcSDR* genes and to identify conserved motifs, respectively. TBtools II [47] was applied for mapping *CcSDR* genes onto chromosomes, identifying *SDR* paralogous genes within *C. camphora*, and finding *SDR* orthologous genes among *C. camphora*, *P. trichocarpa*, and *O. sativa*. In this study, the conditions for gene tandem duplication were set as follows: (1) the alignment coverage of the two homologous genes was greater than 70% (the shorter gene relative to the longer gene), and the sequence similarity was greater than 70%; and (2) the distance between the two genes on the chromosome was less than 100 kb.

The multiple sequence alignment of the CcSDR proteins was performed using the ClustalW algorithm (http://www.clustal.org/clustal2/) and refined by TrimAL [48]. Phylogenetic analysis of CcSDRs was conducted using MEGA 11.0.13 (https://www.megasoftware.net/) with the maximum-likelihood (ML) method with 1000 bootstrap replicates and visualized using the online tool ChiPlot 2.6.1 (https://www.chiplot.online).

### 4.3. Gene Expression Analysis

Clean data from RNA-seq of forty-five samples were processed using in-house Perl scripts. Only reads with a perfect match or one mismatch were mapped to the *C. camphora* genome assembly using TopHat 2.1.1 (http;//tophat.cbcb.umd.edu/). Mapped reads were then assembled to yield transcripts by Cufflinks 1.3.0 (http://cole-trapnell-lab.github.io/cufflinks/install/) and StringTie 2.2.0 (http://ccb.jhu.edu/software/stringtie/). Gene expression levels were estimated by FPKM values. The thresholds for significantly differential expression were set at a *p*-value < 0.01 and |log2 (fold change)| ≥ 1.0. The FPKM values of differentially expressed *CcSDR* genes were first transformed into log2 values and then normalized among samples using the scale function from an R package. TBtools II [47] was used to construct a heat map to visualize the results from the clustering analysis of differentially expressed *CcSDR* genes. Mfuzz 2.64.0 (http://mfuzz.sysbiolab.eu/) was applied to display the results from the clustering analysis of differentially expressed *CcSDR* genes across four developmental stages of leaves and five developmental stages of fruits.

### 4.4. LEOs Extraction and Component Identification

A total of 600 g fresh leaves of twelve plants (three plants per chemotype) from four chemotypes was sampled in September 2023. The LEOs of each plant were extracted by steam distillation (three replicates) and the average yield was calculated (volume/weight, *v*/*w*). The components of the LEOs were identified by GC-MS following Qiu et al. [49].

### 4.5. Screening, Cloning, and Biochemical Characterization of CcEGS

Firstly, a phylogenetic tree was constructed using 22 CcSDR460A proteins and biochemically characterized proteins of the PIP reductase family from other plants with the maximum-likelihood (ML) method. Based on the topological structure, CcSDR460A proteins both in the EGS/IGS clade and the PCBER/EGS/IGS clade were preliminarily identified as candidate CcEGSs. The expression levels of these candidate *CcEGSs* in the leaves of four chemotypes were then evaluated by qRT-PCR analysis. Total RNA extraction and purification were performed as previously described. First-strand cDNA was synthesized using the PrimeScript™RT reagent Kit (Takara, Shiga, Japan). Primers for qRT-PCR analysis were designed using Premier 5.0 software and are listed in Appendix A. *CcActin* was employed as the reference gene. The TB Green^®^ Premix Ex Taq™ kit (Takara) was used for the qRT-PCR experiment, with a total reaction volume of 20 µL and cycling conditions as follows: 95 °C for 30 s; 95 °C for 5 s, 60 °C for 30 s, for 40 cycles; and 72 °C for 30 s. The 2^−ΔΔct^ method [50] was used to calculate expression levels, and the *CcSDR* gene with significantly higher expression in the methyl eugenol type relative to other chemotypes was confirmed as a candidate *CcEGS*. Statistical differences were determined by one-way ANOVA using SPSS 26.0 software.

The ORF sequence of *CcEGS* was obtained by PCR amplification. Specific primers for *CcEGS* were designed based on the genome of the camphor tree (Appendix A) and used to amplify the CDS sequences with cDNAs from four chemotypes as templates using the TaKaRa LA Taq kit (Takara). The PCR conditions were as follows: 95 °C for 3 min; 94 °C for 30 s, 55 °C for 30 s, 72 °C for 1 min, for 35 cycles; and 72 °C for 5 min. The PCR product was detected by 1% agarose gel electrophoresis, and the target fragment was purified using a TIANgel Purification Kit (TianGen Biotech, Beijing, China).

*E. coli* feeding assays were performed to characterize the function of CcEGS. The ORF region of *CcEGS* was cloned into the *pET28a* vector using the In-Fusion Cloning Kit exv01 (Biogel GeneTech, Hangzhou, China) (Appendix A). Both the recombinant plasmid *CcEGS-pET28a* and the control plasmid *pET28a* were transformed into *E. coli* BL21 cells. Luria–Bertani (LB) liquid cultures of *E. coli* harboring *CcEGS-pET28a* and *pET28a* were induced with 0.5 mM isopropyl β-D-1-thiogalactopyranoside (IPTG) and grown at 24 °C for 16 h, with coniferyl alcohol (final concentration of 100 µg/mL) added as a substrate. Cells were removed by centrifugation (5000 rpm, 5 min), and the spent medium was transferred to fresh tubes for metabolic product analysis. Hexane (10 mL) was added to the spent medium, vortexed for 5 min, and centrifuged to separate the phases. The hexane layers were concentrated to 100 µL, and 10 µL was used for GC-MS analysis.

### 4.6. Subcellular Localization of CcEGS1

We initially predicted the subcellular location of CcEGS using WoLF PSORT 0.2 (https://psort.hgc.jp/) and Softberry ProtComp 9.0 (https://psort.hgc.jp/). The ORF of *CcEGS* was amplified with primers listed in Appendix A and cloned into the *pCAMBIA1300-GFP/N* vector using the In-Fusion Cloning Kit exv08 (Biogel GeneTech, Hangzhou, China) to produce the recombinant plasmid *pCAMBIA1300-GFP-CcEGS*. This recombinant plasmid contained the GFP in-frame at the N-terminus, driven by the 35S promoter. Subsequently, both the recombinant plasmid *pCAMBIA1300-GFP-CcEGS* and the empty *pCAMBIA1300-GFP* were transferred into *A. tumefaciens* strains GV3101 using the conventional freezing-thawing method. The cells were spread on LB solid medium and cultured for 2–3 days at 28 °C. Positive cells were cultured in 100 mL LB liquid medium overnight at 28 °C. After centrifugation and rinsing, the cells were resuspended in infiltration buffer (10 mM MgCl_2,_ 10 mM 2-Morpholinoethanesulphonic acid, and 150 μM Acetosyringone) and incubated for 3 h at room temperature in the dark. The *Agrobacterium* suspension harboring *pCAMBIA1300-GFP-CcEGS* and empty pCAMBIA1300-GFP was transiently injected into the abaxial epidermis of leaves from six-week-old *N. benthamiana*. After 72 h of infection, GFP fluorescence was observed under an Olympus Fv3000 laser scanning confocal microscope (Olympus Corporation, Tokyo, Japan) with excitation and emission wave lengths of 484 nm and 507 nm, respectively.

## Figures and Tables

**Figure 1 ijms-25-10084-f001:**
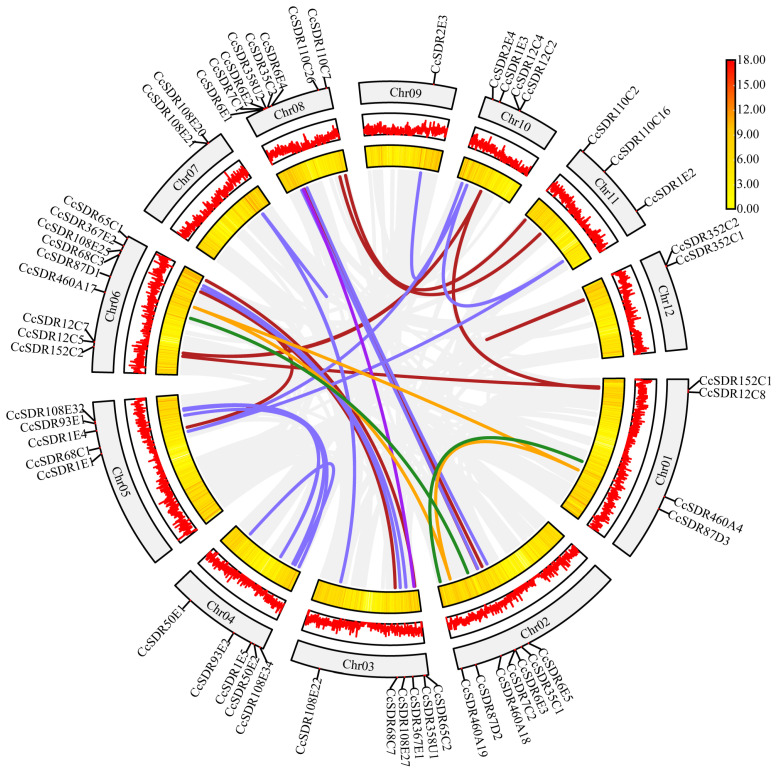
Paralogous genes of the *SDR* gene superfamily arising from segmental duplications in *Cinnamomum camphora*. Red lines represent duplicated gene pairs from the C type; green lines represent duplicated gene pairs from the A type; purple lines represent duplicated gene pairs from the U type; light slate blue lines represent duplicated gene pairs from the E type; and orange lines represent duplicated gene pairs from the D type.

**Figure 2 ijms-25-10084-f002:**
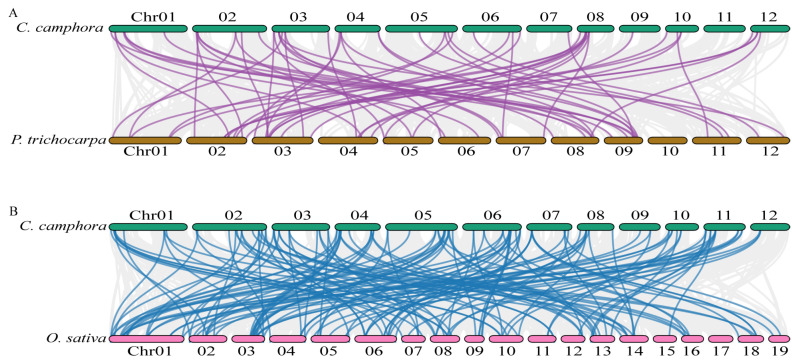
Synteny analysis of *SDR* orthologous genes in the genomes between *C. camphora* and other plants. (**A**) *SDR* orthologous genes in *C. camphora* and *Populus trichocarpa* are highlighted by purple lines; (**B**) *SDR* orthologous genes in *C. camphora* and *Oryza sativa* are highlighted by dark cyan lines. The synteny blocks are shown in gray lines.

**Figure 3 ijms-25-10084-f003:**
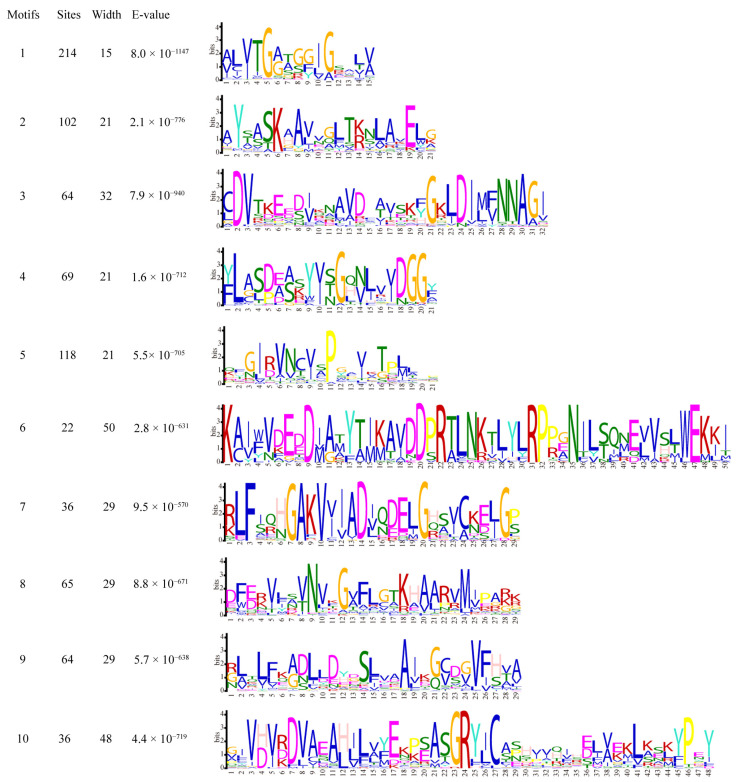
Identification of ten motifs within the *CcSDR* gene superfamily.

**Figure 4 ijms-25-10084-f004:**
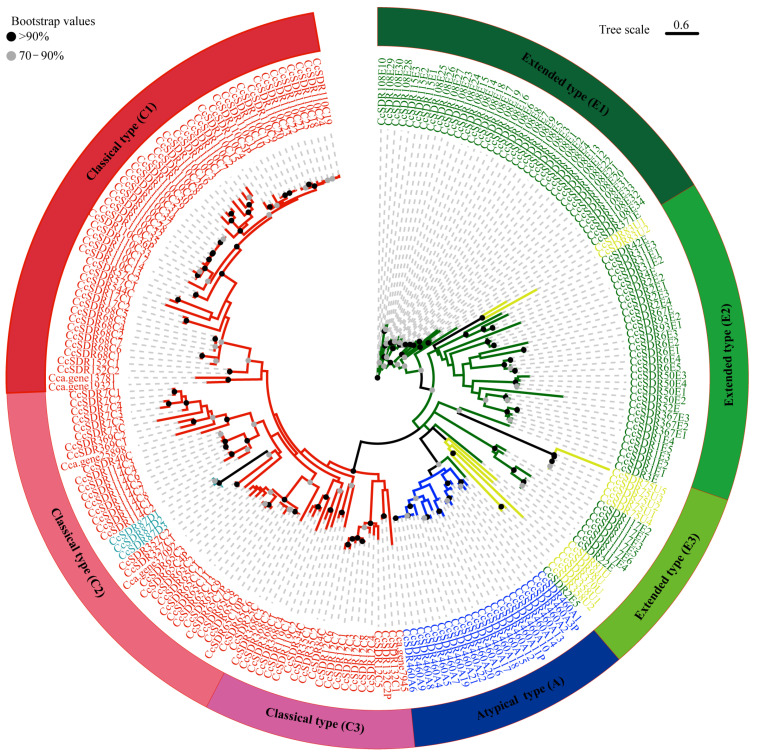
Phylogenetic tree analysis of SDR proteins from *C. camphora*. The CcSDR proteins were divided into three folds based on the clustering of the protein sequence. The proteins from the A, C, D, E, and U types are presented in blue, red, light blue, green, and light green, respectively.

**Figure 5 ijms-25-10084-f005:**
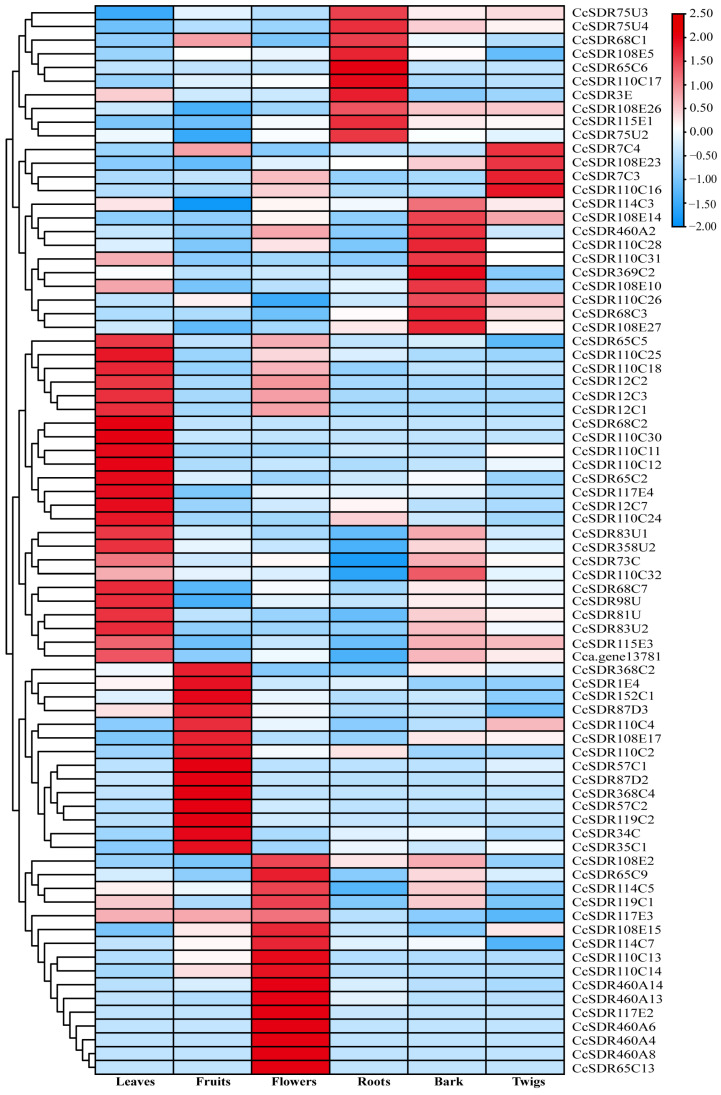
*CcSDR* genes with preferential expression in the leaves, fruits, flowers, roots, bark, and twigs of *C. camphora*. The expression levels in the heat map were adjusted based on the log2 transformation of FPKM values and subsequent normalization.

**Figure 6 ijms-25-10084-f006:**
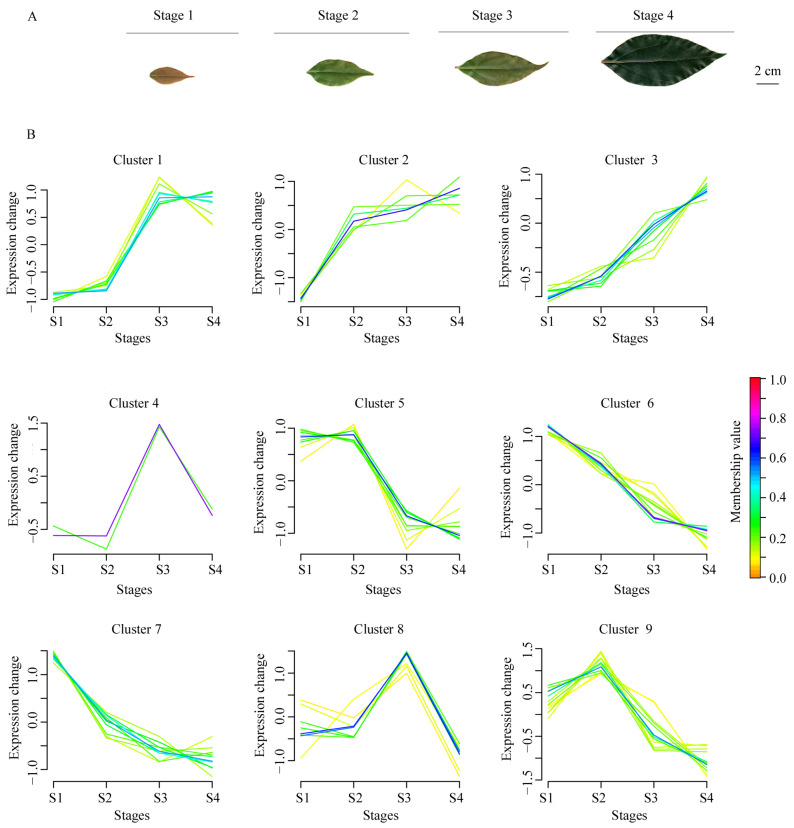
Transcriptional dynamics of *CcSDR* genes in leaves across four developmental stages. (**A**) Samples representing four stages of leaf development were analyzed using RNA sequencing. (**B**) Cluster analysis of DEGs in leaves across four developmental stages.

**Figure 7 ijms-25-10084-f007:**
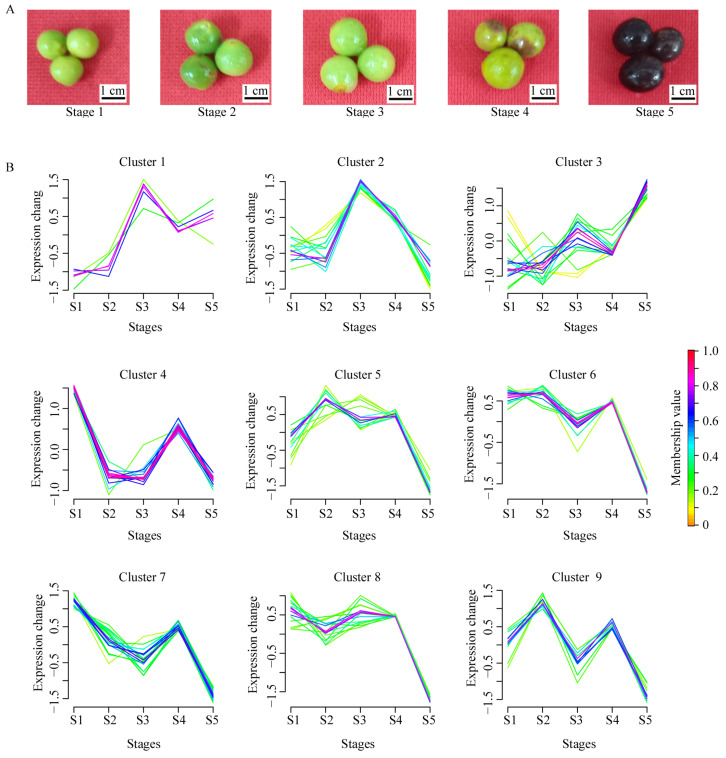
Expression variation of *CcSDR* genes in fruit across five developmental stages. (**A**) Fruits at each of the five developmental stages were collected and underwent RNA-seq. (**B**) Cluster analysis of DGEs in fruits across five developmental stages.

**Figure 8 ijms-25-10084-f008:**
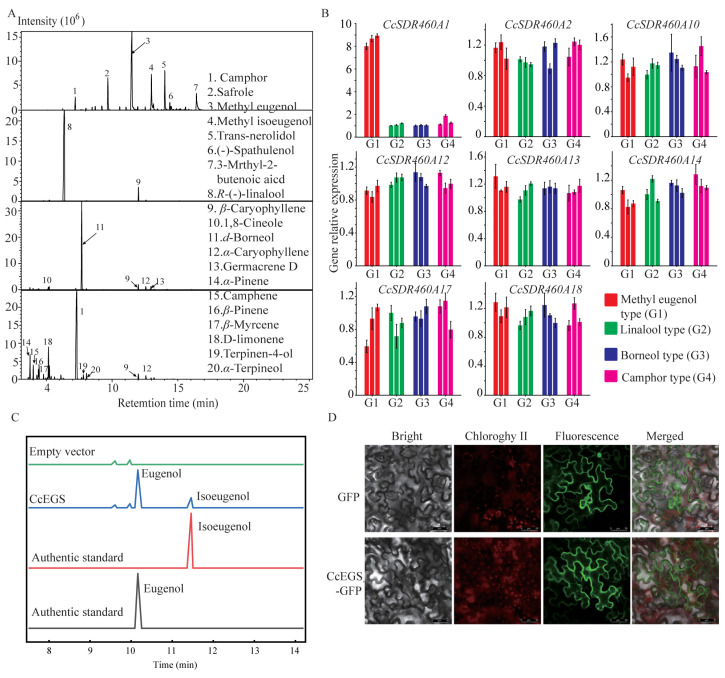
Characterization of eugenol synthase (EGS) from *C. camphora*. (**A**) Fresh leaf essential oil profiles of methyleugenol-type, linalool-type, borneol-type, and camphor-type in camphor tree. (**B**) Evaluation and comparison of expression levels of candidate *CcEGSs* in leaves among methyleugenol-type, linalool-type, borneol-type, and camphor-type in camphor tree. (**C**) Gas chromatography–mass spectrometry analysis of products formed in *E. coli* spent medium by CcEGS. (**D**) Subcellular localization of CcEGS in *Nicotiana benthamiana* leaves. The adaxial leaf surface was observed with laser confocal microscopy (Bars = 50 µm). GFP, green fluorescent protein fluorescence image; chlorophyll, chlorophyll autofluorescence image; bright, transmission image; merged, all channels (GFP, chlorophyll, and bright) combination.

## Data Availability

The sequence data of *CcEGS* have been deposited in GenBank under the accession number PQ41632. The RNA sequencing data of five organs (flowers, bark, twigs, roots, and leaves), four developmental stages of leaves have been deposited into CNGB Sequence Archive (CNSA) of China National GeneBank DataBase (CNGBdb) under the Bioproject numbers CNP0005973 and CNP0005950. The RNA sequencing data of five developmental stages of fruits are available from the corresponding author upon request.

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
