# Peer review of "Genome-Wide Analysis and Characterization of the SDR Gene Superfamily in Cinnamomum camphora and Identification of Synthase for Eugenol Biosynthesis"

_ijms, 2024, doi:10.3390/ijms251810084_

Round 1
Reviewer 1 Report
Comments and Suggestions for Authors
1. English writing could benefit from refinement, particularly with regards to tense consistency and the proper italicization of gene names within the text, which necessitate a meticulous review.
2. When conducting interspecific collinearity analysis, it is advisable to incorporate a comparative study between C. camphora and monocotyledonous plants. This approach will facilitate a deeper understanding of the orthologous SDR genes within terrestrial flora and shed light on their conserved roles throughout plant evolution.
3. According to previous literature, the substitution of a single residue can substantially affect the product specificity of phenylpropene synthases. The question arises whether the CsEGS enzyme sequences from various chemotypes—methyl eugenol, linalool, borneol, and camphor—are completely identical? Could there be key polymorphic residues of CsEGS that might affect the composition of the products?
Author Response
Dear reviewer
We would like to thank you for your kind letter and for constructive comments and valuable suggestions on our manuscript (ijms-3174253).These comments and suggestions are all helpful for improving our article. According to your suggestions and comments, we have re-conducted the collinearity analysis, performed transcript sequence analysis, and enhanced the English writing quality in our manuscript. Below you will find our point-by-point responses to your comments or suggestions:
- English writing could benefit from refinement, particularly with regards to tense consistency and the proper italicization of gene names within the text, which necessitate a meticulous review.
Response: Many thanks for your careful suggestions. We are sorry for a lot of minor issues in our original manuscript. The manuscript has undergone further editing to enhance the level of English writing.Tense consistency, proper italicization of gene names, and other minor grammatical errors within our manuscript were revised (highlighted in red).
- When conducting interspecific collinearity analysis, it is advisable to incorporate a comparative study between C. camphora and monocotyledonous plants. This approach will facilitate a deeper understanding of the orthologous SDR genes within terrestrial flora and shed light on their conserved roles throughout plant evolution.
Response: Thank you for pointing this out. We agree with this comment. Therefore, we re-conducted the collinearity analysis between C. camphora and Populus trichocarpa, as well as between C. camphora and Oryza sativa (Lines 172-186 and Figure 2). The result show the orthologous SDR genes between dicotyledonous and monocotyledonous plants primarily participate in sugar and lipid metabolism, which are essential for the survival of terrestrial plants. However, multiple SDR genes potentially involved in secondary metabolism are non-collinear between dicotyledonous and monocotyledonous plants, indicating that SDR-mediated secondary metabolic reactions may have diverged after the evolutionary separation of dicots and monocots.
- According to previous literature, the substitution of a single residue can substantially affect the product specificity of phenylpropene synthases. The question arises whether the CcEGS enzyme sequences from various chemotypes—methyl eugenol, linalool, borneol, and camphor—are completely identical? Could there be key polymorphic residues of CcEGS that might affect the composition of the products?
Response: Thank you for your insightful suggestion. Acting on your recommendation, we have compared the transcript sequences of the CcSDR gene across different chemotypes of camphor trees and observed complete consistency (Lines 347-349). This finding suggests that CcEGS likely regulates the biosynthesis of methyl isoeugenol by transcriptional regulation or other mechanisms, rather than by key polymorphic residues.
We tried our best to correct minor issues and improve the manuscript.We appreciate for reviewer’ warm work earnestly, and hope that the correction will meet approval. Thank you very much for your comments and suggestions.
Reviewer 2 Report
Comments and Suggestions for Authors
Zhang et al perform genome -wide analysis of SDR gene family in cinnamonium camphora and identified gene for eugenol biosynthesis.
The pare is relativley well design and written.
But some corrections are required.
Line 31: I am not sure thses names are very informative in abstracts. Maybe just three family is enough.
It will be importnat to add to introduction few sentences about Cinnamomum camphora: why this plants is important, why eugenol attract your interest.
Line 110: with = plus.
Lines 114- 116: Ia m not sure average can be count in this case. Average can be count if you have a real Gauss-type distribution.
Line 123: „into into“. are these 5 genes from leaf?
Line 132: average 5,05 exons?
Line 217: „up-regulated in specific organs“ ?? maybe have higher mRNA level? Up-regulation means response to some stimuli.
Figure 5: Please consider that organ have mnay different cepp type and expression patterns in futire should be linked with cell type/position. Expression in epìdermis and pericycle in the root, as an exmaple, have a different meaning.
Figure 6: very interesting data, shown an opposite regulation of gene expression of different clustares. What about cluster 4 and 8? To which cell types they are belong? Clusters 5-7 and 9 may related with general chromatin condensation during leaf maturation, indeed.
Figure 7: similar comments: which cell types belong each cluster? Organ cotain so many cell types, each have own regulation/trajectory.
Figure 8: what is the sites of eugenol production? Whichn cell type? Is it measophyll cells of other cell types? If not mesophyll, why did you choose another species like Nicotiana, abd very specific cell?
Comments on the Quality of English Languagemoderate corrections
Author Response
Dear Review
Thank you for your kind letter and for the constructive comments and valuable suggestions on our manuscript (Manuscript ID ijms-3174253). We appreciate the feedback, as it is instrumental in enhancing the quality of our article.We regret that we are currently unable to provide more detailed information on some issues, but these have provided directions for our future research. In response to your suggestions and comments, we have made the following revisions:
Line 31: I am not sure thses names are very informative in abstracts. Maybe just three family is enough. It will be important to add to introduction few sentences about Cinnamomum camphora: why this plants is important, why eugenol attract your interest.
Response: Thank you for pointing this out. We agree with this comment. Therefore, we have refined the abstract (Lines 15-16, 22-23, 27-31) and added keyword (Line 31). The background of the camphor tree has been specifically supplemented in both the abstract (Lines 15-16) and the introduction (Lines 104-109, 110-118). In the introduction, we have also elaborated on the significance of eugenol and the rationale for conducting this study on its biosynthetic mechanism.
Line 110: with = plus.
Response: Many thanks for your careful suggestions. It was revised according to your suggestions in Line 121.
Lines 114- 116: Ia m not sure average can be count in this case. Average can be count if you have a real Gauss-type distribution.
Response: Thank you for pointing out the incorrect usage. In fact, the vast majority of SDR sequences are concentrated between 250-350 amino acids in length, and there are 10 SDR sequences that exceed 500 amino acids in length. Therefore, we have corrected the description of the results in Lines 124-127.
Line 123: „into into“. are these 5 genes from leaf?
Response: Thank you for your meticulous suggestions. We have made revisions according to your suggestions in Line 135. These 5 genes were obtained based on genomic predictions and were also found to be effectively expressed in leaf tissue according to transcriptomic data.
Line 132: average 5,05 exons?
Response: Thank you for for pointing out the incorrect usage. we have corrected the description of the results in Lines 143-145.
Line 217: „up-regulated in specific organs“ ?? maybe have higher mRNA level? Up-regulation means response to some stimuli.
Response: Thank you for pointing out the incorrect usage. The term "up-regulated in specific organs" refers to having a higher mRNA level. We have corrected the description of the results accordingly in Lines 236, 262-263, 433, 536.
Figure 5: Please consider that organ have mnay different cepp type and expression patterns in futire should be linked with cell type/position. Expression in epìdermis and pericycle in the root, as an exmaple, have a different meaning.
Response: Thank you for your suggestion. We concur with this feedback. Indeed, the characterization of SDR functions should be linked to specific cell types or positions. Regrettably, at present, we are unable to provide expression profiles of SDR genes across various cell types, and can only offer reference information at the organ level for other researchers. It is our hope that future single-cell sequencing studies will yield more reliable expression data.
Figure 6: very interesting data, shown an opposite regulation of gene expression of different clustares. What about cluster 4 and 8? To which cell types they are belong? Clusters 5-7 and 9 may related with general chromatin condensation during leaf maturation, indeed.
Response: Thank you for your valuable comments. In light of your suggestions, we have undertaken a thorough reevaluation of the eight SDR genes from cluster 4 and 8. By analogy with their Arabidopsis homologs, it appears that CcSDE2E1, 6E2/3, and 50E3 are likely involved in sugar metabolism and cell wall formation, while CcSDR31E3 may be associated with the regulation of leaf growth. However, we currently do not possess the necessary data to determine the specific cell types from which these genes are expressed.
Figure 7: similar comments: which cell types belong each cluster? Organ cotain so many cell types, each have own regulation/trajectory.
Response: Thank you for your insightful comments. As previously mentioned, we are currently unable to provide further transcriptional information of CcSDR genes regarding specific cell types of fruits.
Figure 8: what is the sites of eugenol production? Whichn cell type? Is it measophyll cells of other cell types? If not mesophyll, why did you choose another species like Nicotiana, abd very specific cell?
Response: Thank you for your insightful comments. In the leaves of the camphor tree, eugenol and other essential oils are produced in oil cells, which are specialized cells distinct from mesophyll cells. Currently, a transient expression system for camphor trees has not been established, so it is necessary to utilize other model plants, such as Nicotiana, to explore the subcellular localization of CcEGS.
We tried our best to correct minor issues and improve the manuscript.We appreciate for reviewer’ warm work earnestly, and hope that the correction will meet approval. Thank you very much for your comments and suggestions.
Round 2
Reviewer 2 Report
Comments and Suggestions for Authors
Thank you! The authors answered all points. Minor polishing in the text are required.
Comments on the Quality of English Languageminor polishing of some sentences.